# Lifestyle Modification and Atrial Fibrillation: Critical Care for Successful Ablation

**DOI:** 10.3390/jcm11092660

**Published:** 2022-05-09

**Authors:** John L. Fitzgerald, Melissa E. Middeldorp, Celine Gallagher, Prashanthan Sanders

**Affiliations:** 1Centre for Heart Rhythm Disorders, University of Adelaide, Adelaide 5000, Australia; john.fitzgerald@adelaide.edu.au (J.L.F.); melissa.middeldorp@adelaide.edu.au (M.E.M.); celine.gallagher@adelaide.edu.au (C.G.); 2Department of Cardiology, Royal Adelaide Hospital, Adelaide 5000, Australia

**Keywords:** atrial fibrillation, risk factor management, catheter ablation, risk factor modification, lifestyle modification

## Abstract

Management of atrial fibrillation (AF) requires a comprehensive approach due to the limited success of medical or procedural approaches in isolation. Multiple modifiable risk factors contribute to the development and progression of the underlying substrate, with a heightened risk of progression evident with inadequate risk factor management. With increased mortality, stroke, heart failure and healthcare utilisation linked to AF, international guidelines now strongly support risk factor modification as a critical pillar of AF care due to evidence demonstrating the efficacy of this approach. Effective lifestyle management is key to arrest and reverse the progression of AF, in addition to increasing the likelihood of freedom from arrhythmia following catheter ablation.

## 1. Introduction

With a rising incidence globally, atrial fibrillation (AF) continues to be the most common sustained cardiac arrhythmia encountered clinically [1,2]. Global data confirms trends of rising health-care utilisation related to AF and clear association with increased all-cause mortality, stroke and heart failure [2,3]. The key pillars of evidence-based AF management, as described in international guidelines including those of the European Society of Cardiology, American Heart Association/American College of Cardiology, the National Heart Foundation of Australia and others include anticoagulation for stroke risk-reduction, rate control, rhythm control and risk factor management [4,5,6]. Risk factors predisposing to AF contribute to changes in left atrial (LA) structure and function through haemodynamic, electrophysiologic and metabolic effects on the atrial myocardium, with inadequate management contributing to increasing AF burden, progression and resultant complications related to AF [7,8,9,10,11,12]. The standard trajectory of atrial fibrillation with progression to persistent, then long-standing persistent and finally permanent AF can be arrested and reversed by risk factor management as part of comprehensive care delivery from the time of AF diagnosis [13,14,15]. In this review the evidence for AF management strategies and mechanisms of influence of major modifiable risk factors will be discussed. We will then focus on evidence supporting risk factor modification in conjunction with ablation for improving AF outcomes. Finally, we will review practical aspects of delivery of cardiovascular risk factor modification to the AF population.

## 2. Rhythm Control for Atrial Fibrillation

Management of atrial fibrillation with rhythm control has been hampered by low efficacy and deleterious side-effect profiles of anti-arrhythmic medication, leading to recommendations for prioritisation of rhythm control only in patients with ongoing symptoms despite efforts at adequate rate control [16,17]. The CABANA clinical trial of catheter ablation compared with anti-arrhythmic drug therapy in 2204 patients with a mean of 4 years’ follow-up failed to show a difference in the composite primary outcome of death, disabling stroke, serious bleeding, or cardiac arrest, though there were lower than expected event rates and significant rates of between-group cross-over tempering interpretation of these results [18]. Recently, the EAST-AFNET 4 trial randomised 2789 patients with AF duration ≤ 1 year to early rhythm control compared with usual care, where rhythm control was reserved for symptom management [19]. This trial showed that patients randomised to early rhythm control (19% had AF ablation) had a lower risk of the composite outcome of death from cardiovascular causes, stroke, hospitalisation with worsening heart failure or acute coronary syndrome (hazard ratio 0.79; 96% CI 0.66 to 0.97, *p* = 0.005), and also had lower risk of individual secondary endpoints or death from cardiovascular causes and stroke. Event rates were low overall in the trial, likely reflecting contemporary use of medical management for cardiovascular comorbidities and risk factors. Overall, this trial supports the use of an early rhythm control strategy to reduce adverse clinical outcomes related to AF.

The success of atrial fibrillation ablation demonstrates attrition over time. In a meta-analysis of long-term success-rates, 80% success at 5 years was demonstrated, allowing for multiple procedures (though this was only 50% after a single procedure) [20]. In a single-centre study of outcome in 255 patients at 10 years post ablation (43% paroxysmal), 52% remained arrhythmia free and 10% progressed to permanent AF [21]. Multivariate predictors for freedom from AF recurrence were no rise in blood pressure, BMI or fasting glucose over follow-up, highlighting the importance of cardiovascular risk factor management in this population.

There is evidence for progression of atrial cardiomyopathy, and therefore substrate for AF, despite successful catheter ablation. Despite demonstrated reduction in left atrial size following catheter ablation for AF, a contact-mapping study demonstrated atrial electroanatomic abnormalities were greater in AF patients than a control group at baseline and progressively worsened over 10 ± 14 months following AF ablation [22]. This indicates there is a progressive atrial substrate in AF which is not addressed by ablation alone. Additional substrate ablation approaches at the time of initial AF ablation compared with pulmonary vein isolation alone have so far failed to improve post-ablation outcomes, suggesting these do not significantly impact the atrial changes occurring progressively post-ablation. Risk factors for the development of AF have been associated in various studies with development and progression of this AF substrate.

## 3. Modifiable Risk Factors for Atrial Fibrillation

Obesity, physical inactivity, hypertension, obstructive sleep apnoea, diabetes mellitus, alcohol consumption and smoking have all been implicated in increasing the risk of AF development [23,24,25,26,27,28,29,30,31]. Mechanisms for these risk factors increasing atrial fibrillation have been explored in animal and human studies and include increased LA size, pressure, blood volume, increased central blood volume and systemic vascular resistance, left ventricular hypertrophy and stiffening, reduced diastolic filling, adverse changes in the renin-angiotensin-aldosterone system, increased atrial fibrosis, inflammatory and prothrombotic changes and alterations in left atrial voltage, refractoriness and conduction velocities [7,8,9,10,11,12,32,33,34,35,36,37]. Modifiable cardiovascular risk factors play a critical role in the development of AF (See Table 1 for a summary of risk factors for AF). Similarly, poor control of these risk factors is associated with a heightened risk of AF recurrence post-ablation. The presence of individual risk factors and poor control of these at the time of AF ablation has been associated with poorer outcomes in multiple studies.

## 4. Risk Factor Modification before and in Conjunction with AF Ablation

Several studies have highlighted the effectiveness of comprehensive cardiovascular risk factor management in reducing AF burden and progression [13,14,15]. Whilst some studies of isolated risk factor modification, including alcohol abstinence and bariatric surgery for morbid obesity have demonstrated improved arrhythmia freedom [50,51], a comprehensive approach is likely to be more sustainable and translatable, given the interdependence and co-existence of modifiable cardiovascular risk factors. The cost-effectiveness of comprehensive risk factor management delivery has also been demonstrated [52].

Management of isolated risk factors, such as obesity, in the context of AF ablation have demonstrated variable success. Some studies have shown improvement in outcomes, particularly with treatment of severe hypertension and OSA, but to a lesser degree than that of comprehensive risk factor management programs (Figure 1).

## 5. Obesity and Ablation

In patients undergoing AF ablation, increased BMI is associated with increased AF recurrence. Although earlier studies showed somewhat conflicting outcomes [53,54], the greatest risk of AF recurrence occurs with BMI of >30 kg/m^2^ [55]. In a study of 771 paroxysmal AF patients undergoing AF ablation with pulmonary vein isolation, the recurrence of AF at up to 1 year progressively increased with higher BMI categories, and was up to 58% if BMI was >40 kg/m^2^ [56]. Similarly, each five-unit increase in BMI has been associated with a 13% increase in likelihood of AF recurrence post ablation [23].

The impact of a weight loss intervention on AF recurrence after catheter ablation was evaluated in the recently published SORT-AF randomised trial [57]. In this multi-centre study, risk factors were assessed in both groups with a sleep study undertaken in all patients. Hypertension and new diabetes were managed according to guidelines pre ablation. The intervention group participated in a structured weight-loss program with twice monthly medical attendance, regular nutrition advice and assistance with physical training for 6 months. A specialised obesity department with endocrinology oversight was the setting of weight-loss guidance following guidelines from the Medical Society for Treatment of Morbid Obesity. The intervention group achieved a BMI reduction from a mean of 34.9 to 33.4 kg/m^2^ with on average 4.6 kg weight loss (3.9% initial body weight) and 33% non-compliance demonstrated at 12 months, whereas the control group lost an average of 2 kg from BMI 34.8 to 34.5 kg/m^2^. There was no significant difference between groups in AF recurrence (as detected by implantable loop recorder) in the primary intention to treat analysis, but ancillary analysis showed reduced AF burden was associated with BMI reduction, where this was achieved, particularly in participants with persistent AF. The small degree of weight-loss achieved is likely to account for the outcomes observed in this study.

An observational study of weight loss in 90 patients with long-standing persistent AF showed improved quality of life, but no improvement in symptom severity or AF recurrence at 1 year of follow-up following a single AF ablation procedure which involved pulmonary vein antral isolation, posterior wall isolation and non-pulmonary vein trigger ablation [58]. In 58 patients who achieved a significant 24.9 (IQR 19.1–56.7) kg weight loss from a mean BMI of 38 ± 4, there was no significant difference in freedom from AF at 1 year (63.8%) compared to a control group of 32 patients with similar BMI of 37 ± 5 who achieved no significant weight loss, with 1-year freedom from AF in 59.3%. Although alcohol reduction and smoking cessation were strongly encouraged in both groups, dietician-guided calorie reduction and encouragement to exercise with a daily diary kept of these were the main interventions studied. Reasons proposed for this failure of effect of weight loss are the more advanced AF substrate of long-standing persistent AF, likely with less reversibility potential compared with paroxysmal AF and perhaps closer follow-up for increased AF detection. It is of note that only a single procedure was performed in this study, whereas it is known that multiple procedures are often required for successful elimination of AF, particularly in the setting of persistent or long-standing persistent AF [59].

## 6. Physical Inactivity and Ablation

Data for the effects of physical inactivity in isolation following AF ablation are limited, though the effect of the level of cardio-respiratory fitness (a surrogate for physical activity) at the time of AF ablation has been studied. In a single-centre study of 591 patients with a mean follow-up of 32 months, lower cardio-respiratory fitness, as assessed on exercise stress test within 12 months pre-ablation, was associated with increased AF recurrence, with 79% in the low (<85% predicted) versus 28% in the high (>100% predicted) cardiorespiratory fitness groups experiencing arrhythmia recurrence post AF ablation, respectively [60].

A cardiac rehabilitation model of intervention after AF ablation was tested in the CopenHeart RFA trial [61]. In this study, 210 patients (72% paroxysmal AF) undergoing AF ablation were randomised to cardiac rehabilitation with usual care or usual care alone. Cardiac rehabilitation involved 12 weeks of physical exercise sessions and four psycho-educational consultations. Physical capacity increased in the cardiac rehabilitation group as measured by VO_2_ Max, though there was not a statistical difference in the METs of exercise done regularly by each group, and there was no difference in mental health components of the SF-36 questionnaire between groups. A difference in VO_2_ max was sustained at 12 months in the cardiac rehabilitation group and a lower proportion of patients had high anxiety at 24 months in this group. No difference in hospitalisation or mortality was seen in the rehabilitation group at longer-term follow-up of 24 months [62]. While some fitness effects were long-lasting, the duration of input following ablation was relatively short and not as broadly focussed as a comprehensive risk-factor modification approach.

## 7. Hypertension and Ablation

The effect of hypertension on atrial fibrillation recurrence after ablation is influenced by the level of control of this risk factor. Hypertension associates with increased age, more cardiovascular comorbidities, greater likelihood of persistent AF, and was an independent predictor of increased recurrence of AF post-ablation in earlier studies [63,64,65,66]. A more recent study of 626 patients from 55 centres in the German Ablation Registry showed that AF recurrence rates, freedom from antiarrhythmic medication and repeat ablation were not different with versus without hypertension diagnosis at the time of ablation, though there were more reports of dyspnoea, angina and more re-hospitalisations in those with hypertension [66]. A further multi-centre study of 531 consecutive patients undergoing AF ablation showed again that hypertension itself was not associated with a higher recurrence rate of AF following AF ablation, but poor control of hypertension despite medical therapy pre-procedure did lead to higher recurrence [65].

In a randomised study of renal artery denervation for drug-resistant hypertension in patients referred for pulmonary vein isolation, a treatment group of 13 patients showed significant BP reduction from 181/97 to 156/87 mmHg compared with no significant change in the control group [67]. In the treatment group, 69% remained free of AF at 12 months compared with only 29% in the control group. In the recent ERADICATE-AF study, patients undergoing ablation for paroxysmal AF with hypertension despite taking at least 1 antihypertensive medication were randomised to renal artery denervation in addition to ablation or ablation alone (PVI) in 154 versus 148 individuals [68]. Freedom from AF, atrial flutter or tachycardia at 12 months was significantly reduced in the renal artery denervation group (72% versus 56%) without any increase in complications. A limitation of both of these renal artery denervation trials is the absence of sham-procedure control for the renal denervation component, a feature that has significantly altered outcomes in recent renal denervation trials for hypertension alone [69].

Aggressive treatment of isolated mild hypertension has not demonstrated reduced likelihood of AF recurrence post AF ablation. In the SMAC-AF randomised open-label trial of aggressive blood-pressure control, targeting < 120 mmHg versus < 140 mmHg in 184 patients undergoing AF ablation, no significant difference in AF recurrence or symptoms were observed at 14 months’ median follow-up. Blood pressures achieved in the two study groups were 123 mmHg vs. 132 mmHg for intervention versus control, respectively [70].

## 8. Diabetes Mellitus and Ablation

Diabetes mellitus is associated with a greater likelihood of AF recurrence post-ablation, particularly in the setting of persistent AF [71,72]. However, a meta-analysis including 1464 patients in earlier studies and review of a large 8175 patient dataset from the German Ablation Registry showed no significant increased atrial arrhythmia recurrence associated with diabetes, although an increased rate of repeat procedures was seen [73,74].

Glycaemic control pre-AF ablation has been studied in a retrospective observational study of 298 patients [75]. A higher glycated haemoglobin (HbA1c) at the time of AF ablation (>9%) was associated with increased recurrence over 25.92 ± 20.26 months (68.75%) compared with an HbA1c < 7% which was associated with 32.4% recurrence. The 12-month pre-ablation trend in HbA1c was a significant predictor of recurrence on multi-variate analysis, with 10% improvement in HbA1c associated with only 2% recurrence, compared with 91% recurrence in patients with a worsening HbA1c trend pre-ablation. This increase in AF recurrence with poorer glycaemic control was also seen in a meta-analysis including data from 1464 patients and presents a strong argument for strict glycaemic control in patients undergoing AF ablation [73].

## 9. Obstructive Sleep Apnoea and Ablation

Initial studies of OSA as a predictor of AF recurrence following AF ablation have yielded conflicting results [54,76,77,78,79]. Meta-analysis of these early studies shows polysomnography-confirmed OSA diagnosis confers independent predictive value for AF recurrence (risk ratio 1.40, 95% CI 1.16–1.68, *p* = 0.0004) but Berlin Questionnaire-based diagnosis does not (risk ratio 1.07, CI 0.91 to 1.27) [80].

Continuous positive airway pressure (CPAP) treatment of known OSA significantly improves outcomes following AF ablation. Meta-analyses of outcomes following ablation for AF in patients with OSA with or without CPAP have incorporated mostly observational studies in the absence of RCTs [76,78,79,81,82,83,84]. One analysis of five observational studies, including 3743 patients showed increased relative risk (1.31, *p* = 0.00) of AF recurrence with a diagnosis of OSA. Untreated OSA was associated with increased recurrence (RR 1.57, *p* = 0.00), whereas treated OSA showed no increase in risk compared to patients without OSA (RR 1.25, *p* = 0.37), with similarity maintained after removal for study heterogeneity [81]. A subsequent meta-analysis comparing CPAP versus no CPAP treatment of OSA following AF ablation showed a 42% decreased risk for AF recurrence in a random effects model, with a meta-regression model showing particular benefit for younger, obese male patients [82].

## 10. Alcohol Consumption and Ablation

Ongoing alcohol consumption has been demonstrated to lead to increased AF recurrence following AF ablation, an effect particularly evident in heavy drinkers. A Japanese study of 1361 patients undergoing AF ablation showed that AF recurrence after the initial procedure was increased in drinkers compared to non-drinkers, though there was no significant difference following repeat procedures [85]. A study of 122 consecutive patients undergoing ablation for paroxysmal AF showed moderate and particularly heavy drinking (defined as >15 g/day for women and >30 g/day for men) was associated with more extensive low-voltage zones and increased AF recurrence [86]. These studies and others are somewhat hampered by reliance on self-report and individual recall of alcohol intake. Objective data using ethyl glucuronide in hair as a long-term alcohol consumption marker, demonstrated an increased likelihood of recurrence of AF in men who met or exceeded a 7 pg/mg cut-off, consistent with ongoing alcohol consumption [87].

## 11. Smoking and Ablation

Little data exists examining the impact of smoking on AF recurrence after ablation. One study of 59 patients (predominantly paroxysmal AF) undergoing pulmonary vein isolation found that ongoing smoking versus non-smoking was independently associated with 43% vs. 14% AF recurrence (*p* < 0.05) at 306 ± 95 days of follow-up. AF recurrence was increased in current and former smokers compared with never smokers [88]. A retrospective study of 201 patients undergoing persistent AF ablation did not show an increased AF recurrence in smokers, but more extensive non-pulmonary vein AF triggers were demonstrated in smokers. Patients with right-atrial non-pulmonary vein triggers seen on mapping at the index procedure had increased recurrence compared to those without RA triggers, pointing to harmful effects of nicotine in promoting these specific non-pulmonary vein triggers and further AF recurrence [89].

## 12. Comprehensive Risk Factor Modification

Comprehensive management of cardiovascular risk factors has been associated with the greatest likelihood of freedom from arrhythmia recurrence post ablation. (Table 2) This was demonstrated in the ARREST-AF study, where in 149 consecutive patients undergoing AF ablation with BMI ≥ 27 kg/m^2^ and ≥1 cardiac risk factor, 61 chose to undertake aggressive risk factor modification whilst 88 declined [90]. In the risk factor modification group blood pressure, lipids (as guided by overall cardiovascular risk), blood glucose levels, and sleep apnoea were all intensively managed, with lifestyle interventions to achieve weight loss and target exercise volume, as well as alcohol reduction/cessation and tobacco abstinence. Significant improvements were observed in multiple risk factors including weight, blood pressure management, glycaemic control and lipid profiles. There was a greater likelihood of single procedure and multiple-procedure drug-unassisted AF-free survival in the risk factor managed group with 16% requiring ongoing anti-arrhythmic drugs compared with 42% in the control group at a mean follow-up of 42 months. AF symptoms were also markedly reduced in the treatment group compared to controls. Overall arrhythmia-free survival in the risk factor management group was 87% compared to 17.8% in the control group. These results provide clear evidence for the importance of continuing to address the underlying substrate progression with AF risk factor management, even after attempts at a curative catheter ablation procedure, if optimal AF-free survival and symptom reduction are to be achieved. While there is little to no data on risk factor management following ablation for AF during open cardiothoracic surgical procedures, it is reasonable to assume that the same substrate progression and AF recurrence can be impacted by comprehensive risk factor modification, given the underlying disease process is the same. Figure 2 summarises the effects of AF ablation with or without risk factor management.

## 13. Structure of Risk Factor Modification Clinics

A structured, co-ordinated, and goal-directed approach to AF management was employed in the LEGACY and ARREST-AF trials and is best fitted to managing cardiovascular risk factors in AF, as it is to many other chronic diseases. Enhanced patient compliance and improved clinical outcomes have been demonstrated with specialised AF clinics [94]. Surrounding the individual patient with AF, a team involving the primary care doctor, heart rhythm specialist, specialised nurse, and where applicable, exercise physiologist, sleep physician, endocrinologist/diabetes care team and pharmacist are best placed to gain optimal control of all risk factors and maximise the likelihood of reduced AF burden and recurrence.

The clinic model employed successfully in the above studies involves an independent risk factor management clinic, which is separate and additional to usual medical care. Physicians and specialist nurses deliver this separate clinic with support from other multidisciplinary team members. Due to the demonstrated success of this model, it is strongly supported as a key pillar of AF care in major current international guidelines including those of the European Society of Cardiology, American Heart Association/American College of Cardiology, and others [4,5,6,95].

Critical components of the risk factor clinic are:Patient centred, individually tailored care;A lifestyle journal including regular recordings of blood pressure, exercise and dietary intake, which is regularly reviewed, with feedback provided;Incremental, goal-directed approach;Flexible follow-up, with access to care between appointments;Surveillance and titration of medication and other treatments;Screening of all patients with AF for the presence of OSA and review of any CPAP treatment data to ensure efficacy.

Where access to a specialised AF risk factor management clinic is not available, strong consideration should be given to the establishment of a dedicated clinic where significant volumes of patients with AF are treated. Another potential solution is to harness chronic disease management models or plans, where these are facilitated by local government and funding models to utilise primary care nurses, doctors and allied health to deliver the components of risk-factor management and forge the necessary relationships with a network of other specialists required to deliver optimal AF care.

## 14. Conclusions

The importance of risk factor modification as a critical pillar of AF care has developed a significant evidence base and is essential to reduce the risk of AF burden and progression. Regardless of treatment strategy employed, it is critical to optimise risk factor profiles to manage this condition. Patient engagement and a unified team approach are required to modify obesity, physical inactivity, hypertension, dyslipidaemia, diabetes mellitus, OSA, alcohol and tobacco use, and this approach is cost effective. The strong evidence for improved short and long-outcomes and ongoing increase in the burden of AF on health-care systems globally point towards the urgency of optimising the management of patients with this arrhythmia. Furthermore, in those undertaking ablation for AF, it is critical to optimise risk factor profiles to maximise the likelihood of successful outcomes from this procedure. The widespread implementation and evaluation of risk factor management clinics is critical to stem the growing burden of AF and associated health care impact.

## Figures and Tables

**Figure 1 jcm-11-02660-f001:**
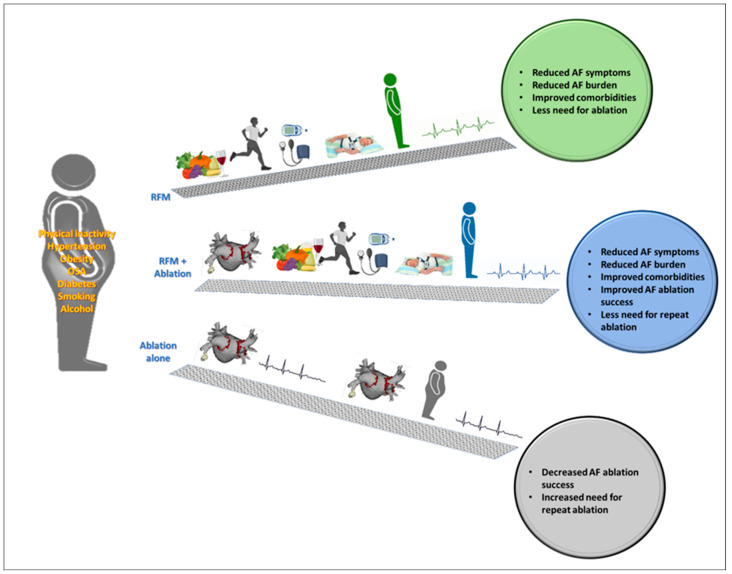
Effects of risk factor management (RFM) for AF including the benefit of undertaking risk factor management as a primary method of care for atrial fibrillation (AF) patients. Benefits are also seen in those who require ablation, with lower benefit for AF ablation alone in those who present with risk factors.

**Figure 2 jcm-11-02660-f002:**
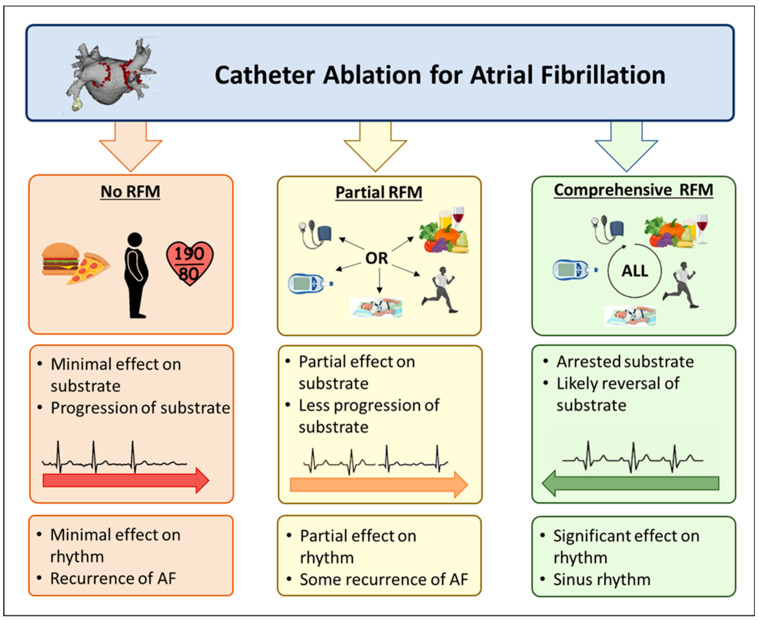
This figure highlights the additive benefit of risk factor management with AF ablation. Whether partial (some risk factors) or comprehensive (all risk factors) a benefit for risk factor management is seen in patients with AF who require ablation.

**Table 1 jcm-11-02660-t001:** Outline of studies which demonstrate the effects of individual risk factors on atrial fibrillation (AF).

Risk Factor	Effect
**Obesity**	Increased LA size, pressure and blood volume, increased central blood volume and systemic vascular resistance, increased epicardial and pericardial fat deposition, conduction slowing [7,8,9,32]. 29% increased risk for developing AF for each 5-point increase in BMI [23].
**Physical Inactivity**	Higher risk of developing AF and association with poorer cardiovascular health and increasing obesity [38]. Up to 28% reduction in the risk of AF was associated with moderate-intensity physical activity in the Cardiovascular Health Study [24].
**Hypertension**	Left ventricular hypertrophy and stiffening, reduced diastolic filling, increased left atrial volumes all associated with hypertensive increased afterload. Left atrial dilatation, adverse electrophysiological changes and circulating hormones such as angiotensin II linked with fibrosis and perpetuation of these changes [10,11,12]. 40 to 50% increased likelihood of developing AF in the Framingham Heart Study [25]. Blood pressure above 130/80 mmHg associated with significantly higher risk of major adverse cardiovascular events in a large cohort of nearly 300,000 patients with known atrial fibrillation [39].
**Obstructive Sleep Apnoea (OSA)**	Repetitive interruption of ventilation via recurrent pharyngeal collapse leads to hypoxaemia-related atrial electrophysiological changes and altered haemodynamics that increase LA pressures, leading to left atrial enlargement [33,34]. Chronic OSA is associated with inflammatory and prothrombotic systemic changes [35]. The risk of developing AF doubles with the presence of OSA [26]. Degree of symptoms is a very poor indicator of the presence or severity of OSA [40]. Considerable night-to-night variability shown in the severity of sleep-disordered breathing [41]. Repeat testing in the case of an unexpectedly negative overnight oximetry study should be strongly considered where clinical suspicion remains high [41].
**Diabetes mellitus**	One-third increase in the risk of developing AF has been independently attributed to diabetes mellitus [27]. In patients with AF, the presence of diabetes has been associated with increased thromboembolism and stroke risk [42,43].
**Alcohol consumption**	Binge-drinking has been associated with increased risk of AF episodes, termed ‘Holiday Heart’ for many years [28]. Chronic alcohol use is associated with increased incidence and burden of AF, and regular moderate alcohol consumption is associated with reduced voltage and slowed conduction on electro-anatomical mapping at the time of AF ablation [29,36]. Chronic alcohol consumption is strongly associated with hypertension, obesity and OSA [29]. While more than seven standard drinks per week was associated with increased risk of developing AF, in a recent study of UK Biobank data, this risk may vary with the type of beverage consumed, with an increased risk of AF seen with reported beer or cider consumption, compared to wine or spirits [30]. Comprehensive review of previous data on alcohol and AF risk suggests that any alcohol at all increases the risk of AF and there is no benefit on AF from light alcohol consumption, which has been seen in studies related to ischaemic heart disease risk [29,44].
**Smoking**	Observational cohort studies, such as the ARIC study, show up to double the risk of developing AF associated with smoking [31]. Mechanisms proposed include increased sympathetic tone, oxidative stress, inflammation and atrial fibrosis. In the presence of AF, smoking increases the risk of thromboembolism and mortality [45].
**Cholesterol levels**	Observational studies have not shown a correlation with lower cholesterol, or particularly LDL and reduced AF, with, in fact a paradoxical relationship where less AF was seen with higher LDL levels [46,47]. The significance of this data is unclear though, and many patients with AF and hypercholesterolaemia have associated increased risk for cardiovascular events related to comorbidities such as hypertension or prior cardiovascular events. Adverse cardiovascular events, rather than AF-risk per se are associated with raised lipids in these instances [48,49].

**Table 2 jcm-11-02660-t002:** Studies demonstrating evidence for risk factor management in patients undergoing AF ablation.

Study	Study Type	Number of Patients	Intervention or Risk Factor Studied *	Population	Change in Risk Factor(s)	Average Follow-Up Duration (Months)	Number of Procedures	Outcomes
Pathak et al., 2014 (ARREST AF) [90]	Cohort	149	Aggressive comprehensive risk factor management	AF patients, BMI > 27 kg/m^2^ 61 RFM 88 standard care	RFM group More weight loss Improved BP control Improved blood sugar control Reduced sleep apnoea	41.9	1.6 ± 0.7	Reduced AF symptom burden (*p* < 0.001) Improved arrhythmia-free survival: 87% arrhythmia free treatment group vs. 17% in control group (*p* < 0.001)
Gessler et al., 2021 (SORT-AF) [57]	Randomised controlled trial	133	6 months of structured weight loss program	AF patients, BMI 34.9 67 weight loss 66 usual care	Weight loss group lost more weight (3.91%)	12	1 17% had >1 ablation	AF burden reduced in both groups post-ablation (*p* < 0.001) but no difference between groups
Mohanty et al., 2018 [58]	Cohort	90	1 year of weight loss intervention	Long-standing persistent AF patients, BMI 38 58 weight loss 32 standard care	Weight loss group Lost 24.9 kg cf control group 0.9 kg (*p* < 0.001)	12	1	No difference in AF symptoms by AFSS Improved physical (*p* = 0.013) and mental (*p* < 0.02) component scores of SF-36 in weight loss group compared to usual care
Pokushalov et al., 2012 [67]	Randomised controlled trial	27	Renal denervation in addition to pulmonary vein isolation versus pulmonary vein isolation alone	AF patients refractory to 2 AADs with drug-resistant hypertension, BMI 28 14 PVI only 13 PVI + renal denervation	Intervention group: BP improved from 181/97 to 156/87	12	1	Intervention group: 69% arrhythmia-free Control group: 29% arrhythmia-free (*p* = 0.033)
Pokushalov et al., 2014 [91]	Meta-analysis of combined data from 2 randomised controlled trials	80	Renal denervation in addition to pulmonary vein isolation versus pulmonary vein isolation alone	AF patients BMI not stated 39 PVI only 41 PVI + renal denervation	Intervention group: BP	12	1	Intervention group: 63% AF-free Control group: 41% AF-free (*p* = 0.014)
Steinberg et al., 2020 (ERADICATE-AF) [68]	Randomised controlled trial	302	Renal denervation in addition to pulmonary vein isolation versus pulmonary vein isolation alone	Paroxysmal AF patients, BMI not stated, 16.8% obese 154 PVI + renal denervation 148 PVI alone	Intervention group: mean BP reduced 150–135 mmHg vs. control group 151–147 mmHg (*p* < 0.001)	12	1	Greater freedom from AF recurrence (72%) in treatment vs. (57%) control group (*p* = 0.006)
Parkash et al., 2017 (SMAC-AF) [70]	Randomised controlled trial	184	Aggressive BP treatment (target <120 mmHg) vs. standard BP treatment (target <140 mmHg)	AF patients, BMI 32, (57% paroxysmal) 92 aggressive BP treatment 92 standard BP treatment	Aggressive BP treatment group mean BP reduced 143–123 mmHg vs. control group 142–135 mmHg (*p* < 0.001)	14	1	Intervention group recurrence of AF/atrial tachycardia/atrial flutter not different to control group (both 61%) (*p* = 0.763)
Fein et al., 2013 [79]	Retrospective cohort	62	Treatment of obstructive sleep apnoea vs. non-treatment	AF patients, BMI 30, 53% persistent AF 32 with OSA on CPAP 30 with OSA no CPAP	Not specified	12	Not specified	Higher atrial tachyarrhythmia-free survival rate with CPAP than without (72% vs. 37%) (*p* = 0.01)
Patel et al., 2010 [84]	Retrospective cohort	3000	Treatment of obstructive sleep apnoea vs. non-treatment	AF patients, BMI 27, 53% paroxysmal 315 with OSA on CPAP 325 with OSA no CPAP	CPAP vs. no CPAP	32	1	Higher AF-free survival rate with CPAP than without (79% vs. 68%) (*p* = 0.001)
Naruse et al., 2013 [83]	Prospective case–control	153	Treatment of obstructive sleep apnoea vs. non-treatment	AF patients, BMI 25, 54% paroxysmal 82 with OSA on CPAP 34 with OSA no CPAP	CPAP vs. no CPAP	19	1	Lower AF recurrence with OSA + CPAP vs. OSA no CPAP (30% vs. 53%) (HR 0.41, CI 0.22–0.76, *p* < 0.01)
Jongnarangsin et al., 2008 [76]	Retrospective cohort	324	Treatment of obstructive sleep apnoea vs. non-treatment	AF patients, BMI 30, 72% paroxysmal 18 with OSA on CPAP 14 with OSA no CPAP	CPAP vs. no CPAP	7	1	Lower AF recurrence with OSA + CPAP vs. OSA no CPAP (50% vs. 71%) (underpowered for this outcome, *p* = 0.289)
Donnellan et al., 2019 [75]	Retrospective cohort	298	Pre-procedure HbA1c control <7% vs. poor control	AF patients with diabetes, BMI 34, 40% paroxysmal *n* = 298	HbA1c controlled to <7% compared to >9%	26	Not specified	AF recurrence lower with HbA1c <7% (32.4%) vs. >9% (69%) (*p* < 0.0001) HbA1c trend in 12 months prior to ablation: 10% improvement showed lower (2%) recurrence vs. HbA1c worsening trend (91%) (*p* < 0.0001)
Donnellan et al., 2019 [92]	Retrospective observational cohort	239	Bariatric surgery vs. no bariatric surgery pre-AF ablation	AF patients, BMI 41, 39% paroxysmal 51 Bariatric surgery vs. 188 no Bariatric surgery	Bariatric surgery vs. no Bariatric surgery	36	1.3	Lower AF recurrence in surgery group vs. non-surgery group (20% vs. 61%) (*p* < 0.0001) Lower repeat procedure requirement with surgery group vs. non-surgery group (12% vs. 41%) (*p* < 0.0001)
Donnellan et al., 2019 [93]	Retrospective observational cohort	255	Bariatric surgery for morbid obesity pre-ablation vs. non-obese	AF patients, BMI 35, 41% paroxysmal 51 Bariatric Surgery BMI 37 vs. 102 no surgery BMI 43 vs. 102 non-obese BMI 25.6	Bariatric surgery vs. no Bariatric surgery vs. non-obese	29	Not specified	Comparable AF recurrence in surgery group (20%) to non-obese group (24.5%), both significantly lower than non-surgery group (55%) (*p* < 0.0001)
Risom et al., 2016 (CopenHeartRFA) [61]	Randomised controlled trial	210	12-weeks of cardiac rehabilitation	AF patients, BMI 28, 72% paroxysmal 105 Cardiac Rehabilitation vs. 105 usual care	Cardiac rehabilitation group improved VO_2_ max at 4 months compared with usual care	6	Not specified	VO_2_ max increased in cardiac rehabilitation group vs. controls, no significant difference in mental health or other SF-36 score components (*p* = 0.20)

* Unless otherwise specified, study intervention groups had intervention + usual care, control group had usual care alone. Usual care did not involve comprehensive risk factor modification unless otherwise specified.

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
