# Peer review of "Lifestyle Modification and Atrial Fibrillation: Critical Care for Successful Ablation"

_jcm, 2022, doi:10.3390/jcm11092660_

Round 1
Reviewer 1 Report
Thank you for allowing me to review the manuscript titled “Lifestyle modification and atrial fibrillation: critical care for successful ablation” with great interest.
This is a very comprehensive review of the risk factor modification after atrial fibrillation therapy.
I have 2 comments.
- Authors should differentiate catheter based vs. surgical ablation when they are referring to various trials
- If possible, authors should attempt to draw Forest plots for the specific risk factors affectig a-fib recurrence after ablation.
Author Response
Reviewer 1: Thank you for your comments. We have made comment on the substrate as requested.
Reviewer 2 Report
This paper by JL Fitzgerald and coworkers reviews comprehensively the potential adverse effect on AF response to catheter ablation provided by the main risk factors belonging to the area of lifestyle. English is fine; the work is well presented and exhaustive concerning the risk factors description , potential adverse effect, based on a thorough literature review.
Since the topic implicate factors leading to a successful ablation , I would personally add a small subsection describing briefly in which case , based on the presence of a given factor, a substrate modification at the index ablation is foreseen , due to a likely diseased left atrium.
Very nice paper.
Author Response
Reviewer 2:
- We have now included a comment on surgical ablation.
- Unfortunately it is not possible to provide relative risks of each risk factor within the scope of this review article.
Round 2
Reviewer 1 Report
I thank authors for taking my comments into consideration.